# Investigating the Relationship Between Ultra-Processed Food Consumption and Academic Performance in the Adolescent Population: The EHDLA Study

**DOI:** 10.3390/nu17030524

**Published:** 2025-01-31

**Authors:** José Francisco López-Gil, Emily Cisneros-Vásquez, Jorge Olivares-Arancibia, Rodrigo Yañéz-Sepúlveda, Héctor Gutiérrez-Espinoza

**Affiliations:** 1One Health Research Group, Universidad de Las Américas, Quito 170124, Ecuador; josefranciscolopezgil@gmail.com; 2Department of Communication and Education, Universidad Loyola Andalucía, 41704 Seville, Spain; 3AFySE Group, Research in Physical Activity and School Health, School of Physical Education, Faculty of Education, Universidad de las Américas, Santiago 7500000, Chile; jorge.olivares.ar@gmail.com; 4Faculty Education and Social Sciences, Universidad Andres Bello, Viña del Mar 2520000, Chile; rodrigo.yanez.s@unab.cl; 5Faculty of Education, Universidad Autónoma de Chile, Santiago 7500912, Chile; kinehector@gmail.com

**Keywords:** healthy eating habits, NOVA classification, nutrition, school performance, youth

## Abstract

**Background:** Previous studies have tested the link between diet quality and academic performance in the young population. However, no study has analyzed the specific relationship between ultra-processed food (UPF) consumption and academic performance in adolescents. The aim of the present study was to test the link of UPF consumption with academic performance in a sample of adolescents from Spain. **Methods:** This secondary cross-sectional analysis utilized information from 788 youths aged 12–17 participating in the Eating Healthy and Daily Life Activities study. The sample comprised 44.7% boys and 55.3% girls, with a median age of 14.0 years (interquartile range [IQR]: 13.0 to 15.0). The UPF consumption was measured through a self-completed food frequency survey. Academic performance was determined using end-of-year academic records provided by each educational institution. To examine the relationships between these variables, generalized linear models were employed. The models were adjusted for factors including sex, age, socioeconomic status, conduct, physical activity, sleep duration, body mass index, and sedentary behavior. **Results:** Significant dose–response associations between UPF consumption and all the different academic performance indicators, showing that higher UPF consumption is consistently associated with poorer academic performance (*p* < 0.001 for all). Higher daily UPF servings were associated with lower adjusted marginal means for grade point average, language, maths, and English. Furthermore, adolescents in the highest UPF tertile had a grade point average of 5.6 compared to 6.6 in the lowest tertile, with similar patterns being observed for language (6.0 vs. 7.0), maths (5.2 vs. 6.2), and English (5.7 vs. 6.6). **Conclusions:** Our study identifies a negative association between UPF consumption and academic performance in adolescents, highlighting it as a modifiable factor that could impact academic outcomes. Adolescents with higher UPF consumption exhibited consistently lower grades across various academic indicators, emphasizing the importance of dietary quality during this critical developmental period.

## 1. Introduction

Adolescence is marked by substantial changes in physical, mental, and social aspects that shape how individuals interact, make choices, think, and experience emotions [1]. Notably, adolescence represents a crucial period for brain development that is characterized by the refinement of synapses, the formation of myelin sheaths, and the establishment of neural connections, particularly in the prefrontal cortex [2,3]. This neurological growth is accompanied by the gradual acquisition of progressively refined cognitive abilities which have a reciprocal relationship with academic performance [4].

The importance of academic performance for adolescents cannot be overstated, as it significantly impacts their growth and future opportunities. For example, one’s academic achievements could influence their educational trajectory and career prospects [5]. Moreover, involvement in educational activities, exploring various disciplines, and expanding one’s knowledge base contribute to the development of analytical thinking, problem-solving capabilities, and cognitive abilities [5]. Successfully meeting academic targets promotes feelings of accomplishment, self-worth, and assurance [6], potentially leading to improved psychological well-being [7]. However, despite these advantages, numerous factors could affect academic performance, including external influences (such as socioeconomic background [8]), personal characteristics (like sedentary lifestyle [9], physical activity [9,10], sleep patterns [11], nutritional choices [12], and weight status [13]), and educational environment (for instance, the type of educational institution [14]).

In the realm of official reports and nutrition research, there is growing recognition that ultra-processed foods (UPFs) are detrimental to human health [15,16,17,18]. UPFs are defined as industrial formulations primarily composed of food-derived substances, additives, and synthetic ingredients [19]. These products are engineered to be convenient, shelf-stable, and highly palatable, but they often contain excessive amounts of salt, sugar, and fat, while offering minimal nutritional benefits, such as in the case of snack foods, sugar-sweetened drinks, ready-to-eat meals, or fast food [20]. The negative impact of UPFs on dietary quality and their association with increased health risks is becoming increasingly apparent [21,22]. From 1990 to 2010, unhealthy food consumption rose globally, with variations being observed across different regions and countries [23]. UPFs constitute a significant portion of diets worldwide, accounting for 20% to over 60% of total energy intake, based on the nation and age group [17]. Despite this prevalence, UPFs have not received adequate attention in public health initiatives [20]. To address this oversight, it is essential to establish a clear link between UPF consumption and cognitive outcomes through robust scientific evidence.

The previous literature has tested the link between diet quality and academic performance in the young population [12,24]. For instance, a systematic review by Burrows et al. [24] pointed out that consistent breakfast consumption, nutrient-poor foods, lower intake of energy-dense foods, and overall diet quality are moderately associated with higher academic performance in the young population. Furthermore, a systematic review and meta-analysis by López-Gil et al. [12] reported a significant association of adherence to the Mediterranean diet with academic performance among children and adolescents. Specifically, some studies have assessed the link between UPFs and cognitive outcomes in the young population [25,26] with contradictory findings. The study by Liu et al. [25] indicated that frequent consumption of individual UPFs (sweet bakery products, candies) and global UPF consumption were related to lower Verbal Comprehension Index scores (a marker of the cognitive function) in children aged 4–7 years from China. Conversely, another study by dos Santos et al. [26] found that no association was identified between UPF consumption and cognitive functioning in low-income adolescents from Brazil.

Although evidence highlights the connection between diet quality and both cognitive and educational outcomes, the specific association between UPF consumption and academic performance in adolescents remains unexplored. This research gap is particularly significant as adolescence represents a critical period for cognitive development and the formation of long-term dietary habits [3]. Understanding this relationship is increasingly urgent given the rising prevalence of UPF consumption and its potential consequences for both health and education [21,22]. In the Spanish context, where dietary patterns are shifting and academic challenges among adolescents are well documented, addressing this issue is especially pertinent [27,28,29]. Therefore, the present study aims to examine the association between UPF consumption and academic performance indicators, focusing on specific subjects such as language, maths, and English, as well as overall grade point averages, in a sample of Spanish adolescents. The findings are expected to contribute to the design of evidence-based interventions and policies that promote healthier eating habits and enhance educational outcomes within this vulnerable population.

## 2. Materials and Methods

### 2.1. Participants and Study Design

This study conducts a secondary cross-sectional analysis using data from the Eating Healthy and Daily Life Activities (EHDLA) study, whose detailed methodology has been previously outlined [30]. The participants were Spanish adolescents aged 12–17 who were enrolled in three secondary schools in the *Valle de Ricote*, Region of Murcia (Spain). Data collection took place during the 2021/2022 school year. The final sample consisted of 788 adolescents (44.7% boys), including all participants with complete data on the variables of interest for this study. Participation required written consent from parents or guardians of the selected teens. They received an informative document explaining the study’s objectives and planned assessments and surveys. The adolescents were also asked to provide their own consent to participate.

The study received ethical approval from multiple institutions. The University of Murcia’s Bioethics Committee granted approval (ID: 2218/2018), as did the Ethics Committee of the Albacete University Hospital Complex and the Albacete Integrated Care Management (ID: 2021-85). The study was also conducted according to the Helsinki Declaration guidelines.

### 2.2. Procedures

#### 2.2.1. Ultra-Processed Food Consumption (Independent Variable)

A self-administered food frequency questionnaire (FFQ), previously validated for the Spanish population [31], was utilized to assess food consumption and energy and nutrient intake. The FFQ comprised 45 items categorized into 12 food groups: (a) red and processed meat, (b) poultry, fish, and eggs, (c) fruits (including preserved fruit), (d) vegetables (salads and other vegetables), (e) dairy products, (f) salted cereals (breakfast cereals, bread, pasta, and rice), (g) sweet cereals (biscuits, pastries), (h) legumes, (i) nuts, (j) sweets (sugar and chocolates), (k) sweetened beverages, and (l) alcoholic drinks. Adolescents indicated their weekly or monthly consumption of these foods, which was then used to calculate the average weekly portion for each group.

The NOVA system [19] was employed to classify UPFs into four categories based on the extent and purpose of industrial processing: (1) unprocessed or minimally processed foods, (2) processed culinary ingredients, (3) processed foods, and (4) UPF and drink products. The classification of UPFs in this study was performed using an *ad hoc* methodology, consistent with previous research. Specifically, the UPF groups were defined following the approach employed in the *Seguimiento Universidad de Navarra* (SUN) cohort study [32]. Moreover, various UPF groups were analyzed separately to provide a more nuanced understanding of their potential impact (see Appendix A).

#### 2.2.2. Academic Performance (Dependent Variable)

The assessment of academic performance was conducted using school records provided by each educational institution at the conclusion of the academic year. Performance was measured through the grade point average (GPA), which encompassed the mean of all academic subject scores for each student (typically spanning 9 to 11 courses), as well as individual evaluations in language, mathematics, and English as a foreign language.

#### 2.2.3. Covariates

The study subjects self-reported their sex and age. Socioeconomic status was assessed using the Family Affluence Scale (FAS-III), which involved tallying responses to six questions about household possessions and facilities (e.g., bedrooms, vehicles, bathrooms, computers, vacations, or dishwashers) [33]. The FAS-III scores ranged from 0 to 13, with higher scores denoting greater affluence. Body mass index (BMI) was determined by dividing the subjects’ weight in kilograms by the square of their height in meters. Total sleep duration was ascertained by asking adolescents about their usual bedtime and wake-up time on weekdays and weekends. The average sleep duration was calculated using the formula [(weekday sleep duration × 5) + (weekend sleep duration × 2)] divided by 7. The Spanish Youth Activity Profile Physical (YAP-S) questionnaire was used to evaluate sedentary behavior and physical activity in young people [34]. This self-reported survey covered a 7-day period and consisted of 15 items grouped into out-of-school activities, school-related activities, and sedentary habits. Energy intake was computed using the previously mentioned FFQ [31].

#### 2.2.4. Justification for the Covariates Selected

The relationship between UPF consumption and academic performance in adolescents is likely influenced by a variety of factors. To account for potential confounding variables, this study included covariates such as sex, age, socioeconomic status, BMI, sleep quality, physical activity, and sedentary behavior. These covariates were selected based on their well-documented associations with academic performance, as evidenced by previous research [12,29,35,36,37,38].

### 2.3. Statistical Analysis

To evaluate the normal distribution of variables, we employed density and quantile-quantile plots and the Shapiro–Wilk test. Due to their non-normal distribution, continuous variables were described using medians and interquartile ranges (IQRs), while categorical variables were expressed as counts (*n*) and percentages (%). Cronbach’s alpha (*α*) was used to assess the internal consistency and reliability of the selected items related to UPF consumption from the FFQ. Since no significant interaction was observed between UPF consumption and sex in relation to academic performance indicators (*p* > 0.05 for all), the analyses combined data from girls and boys. Participants were categorized into tertiles based on their daily UPF serving intake, creating three groups: low, medium, and high consumption. To investigate the relationships between UPF consumption or status and various academic performance indicators in adolescents, generalized linear regression models (GLMs) with robust methods were utilized. These models addressed issues of heteroscedasticity and outliers [39]. GLMs with a Gaussian distribution using the “*SMDM*” method were applied for all continuous outcomes. An *a priori* power analysis was conducted to determine the minimum sample size required to detect a moderate effect size (*R*^2^ = 0.13) in a generalized linear model (GLM) with 9 predictors using a standard significance level (*α* = 0.05 and statistical power (1 − *β* = 0.80). The analysis indicated that a sample size of 114 participants would be required to achieve adequate power. Given that the current study includes 788 participants, the statistical power is well above the recommended threshold, ensuring sufficient sensitivity to detect the hypothesized effects. Additionally, the estimated marginal means (*M*) of the different academic performance indicators, along with their 95% confidence intervals (CI), were calculated. All models were adjusted for covariates such as sex, age, socioeconomic status, sedentary behavior, physical activity, sleep duration, and body mass index. The statistical analyses were performed using R statistical software (version 4.4.0) by the R Core Team in Vienna, Austria, and RStudio (2024.04.1+748) from Posit in Boston, MA, USA. Statistical significance was set at *p* < 0.05.

## 3. Results

Table 1 summarizes the characteristics of the study participants. The median weekly UPF consumption was 4.0 servings (IQR 2.9 to 6.0) and 240.4 g (IQR 157.5 to 385.4). The median GPA, language, maths, and foreign language (i.e., English) scores were 6.3 points (IQR 4.7 to 7.7), 7.0 points (IQR 5.0 to 8.0), 6.0 points (IQR 4.0 to 8.0), and 6.0 points (IQR 5.0 to 8.0), respectively. The internal consistency (*α*) of the UPF items from the FFQ yielded a value of 0.89, indicating high reliability and supporting the coherence of the selected items in measuring UPF consumption.

Figure 1 displays the adjusted marginal means for GPA, language, maths, and foreign language in relation to the rations of UPF consumed. After adjusting for covariates, significant dose–response associations were identified, with greater UPF consumption corresponding to lower GPA (unstandardized beta coefficient [*B*] = −0.58; 95% CI −0.86 to −0.31; *p* < 0.001), language (*B* = −0.56; 95% CI −0.86 to −0.26; *p* < 0.001); maths (*B* = −0.58; 95% CI −0.92 to −0.25; *p* < 0.001), and foreign language (*B* = −0.61; 95% CI −0.90 to −0.32; *p* < 0.001) scores. Furthermore, when analyzing UPF consumption by tertiles (low, medium, and high) in relation to academic performance indicators (GPA, language, maths, and English), a clear trend emerged, showing that higher UPF consumption corresponded to progressively lower performance in all domains (see Appendix A). Notably, significant differences were observed between the low- and high-consumption groups for GPA (*p* = 0.001), language (*p* = 0.001), maths (*p* = 0.011), and English (*p* = 0.003), with a consistent decline in scores being seen as UPF consumption increased. The full results of the GLMs are detailed in Appendix A. An additional analysis examining the association of servings of different UPF groups consumed with academic performance (i.e., grade point average) can be found in Appendix A. These analyses revealed that higher consumption of most of the UPF groups (i.e., fast food, dairy products, beverages, fried foods, and sweets) were significantly associated with lower academic performance (*p* < 0.05 for all), while no significant association was found for sausages.

Table 2 indicates the adjusted marginal means of GPA, language, maths, and foreign language according to the status of the UPFs consumed. Overall, after adjusting for covariates, significant associations were identified. The highest UPF tertile showed the lowest academic performance and the lowest UPF tertile reported the highest academic performance. Adolescents with low UPF consumption showed the highest scores across academic performance indicators, including GPA (6.6, 95% CI 6.4 to 6.9), language (7.0, 95% CI 6.7 to 7.3), maths (6.2, 95% CI 5.9 to 6.5), and English (6.6, 95% CI 6.4 to 6.9). Similarly, those with medium UPF consumption reported slightly lower scores in GPA (6.4, 95% CI 6.1 to 6.7), language (6.7, 95% CI 6.4 to 7.0), maths (6.0, 95% CI 5.7 to 6.4), and English (6.6, 95% CI 6.3 to 6.9). In contrast, adolescents with high UPF consumption displayed the lowest academic performance scores, including GPA (5.6, 95% CI 5.4 to 5.9), language (6.0, 95% CI 5.7 to 6.3), maths (5.2, 95% CI 4.9 to 5.6), and English (5.7, 95% CI 5.5 to 6.0). The differences between high UPF consumption and low or medium UPF consumption were significant for all indicators (*p* < 0.05).

## 4. Discussion

The results of our study highlight that increased consumption of UPFs is associated with lower academic performance, including lower GPAs and reduced performance in language, maths, and English as a foreign language, among the adolescent population studied. These observations are in line with previous research that has demonstrated a relationship between UPF consumption and decreased cognitive function in young people [25,26]. Furthermore, our findings support the existing literature that has established a connection between improved diet quality and enhanced academic performance in youths [12,24]. Although the precise mechanisms through which UPF consumption might affect adolescents’ academic performance are not yet fully understood, several potential explanations could account for these observed relationships.

On the one hand, UPFs are habitually rich in saturated fats, added sugars, and artificial additives while being low in essential nutrients such as vitamins, minerals, and antioxidants [19]. This nutritional imbalance could negatively affect cognitive function and brain development, which are critical during adolescence. For instance, diets high in UPFs are often deficient in omega-3 fatty acids [19], a nutrient linked to improved learning abilities, enhanced cognitive functioning [40,41], and higher relational memory in young populations [42,43]. Moreover, research has demonstrated that supplementary sugars, especially high-fructose corn syrup, negatively impact hippocampal functioning during crucial developmental stages in adolescent rats [44] and American children [45]. In line with this finding, the consumption of a “Western diet” that is high in UPFs such as ice cream, fast food, and chocolate has been linked to decreased neurocognitive performance, particularly in memory processes that depend on hippocampal integrity [46,47,48]. This hypothesis is supported by evidence from both animal models and studies in children. However, caution is warranted when extrapolating findings from animal research to humans. While adolescence is a common developmental stage across mammals, the complexity of the human brain and adolescent behavior is unparalleled [49]. This complexity restricts the direct applicability of animal model findings to human adolescents despite shared neurophysiological features. Further longitudinal and experimental research in humans is needed to better understand these potential effects.

Second, UPFs are often high in sugar [50], leading to rapid fluctuations in blood glucose levels [51]. These variations may negatively affect cognitive performance by reducing alertness and increasing fatigue, thereby hindering the ability to maintain sustained focus on academic tasks [52,53]. In this sense, a comprehensive review and meta-analysis conducted by Mantantzis et al. [54], pointed out that the consumption of carbohydrates significantly reduces alertness within 60 min and increase fatigue within 30 min after ingestion. This short-term energy instability, when consistently experienced, may lead to habitual energy deficits and a reduced capacity to perform academically. While these findings highlight potential mechanisms linking UPF consumption to academic outcomes, further research is needed to confirm these effects in adolescents and explore potential long-term impacts on cognitive functioning.

Third, while our analyses accounted for sleep duration, frequent consumption of UPFs has been associated with poor sleep quality, including insomnia [55] and disrupted sleep patterns [56]. These effects may be attributed to the presence of stimulants, such as caffeine, commonly found in sugary beverages and snacks [57]. Sleep disturbances can significantly impair academic performance by limiting the brain’s capacity to consolidate newly learned information [58] and by reducing critical cognitive functions, including alertness, attention, decision-making, judgment, and overall cognitive efficiency [59]. These findings underscore the potential role of UPF consumption in disrupting sleep and indirectly affecting academic outcomes, warranting further investigation into this connection in adolescent populations.

Fourth, the high consumption of refined sugars and trans fats, commonly found in UPFs, could contribute to chronic low-grade inflammation [60,61]. This systemic inflammation has been linked to impairments in brain function, including reduced synaptic plasticity and memory deficits [62]. Furthermore, inflammation can compromise the integrity of the blood–brain barrier (BBB), a protective structure between the bloodstream and the brain [63]. When the BBB is weakened, inflammatory mediators increase its permeability, allowing immune cells and pro-inflammatory cytokines to infiltrate the brain [64]. This infiltration triggers neuroinflammation, which disrupts neuronal function and is associated with cognitive decline [62]. In adolescents, such disruptions could have significant implications for learning and academic performance. Cognitive functions critical for academic success, such as memory, attention, and problem solving [4], may be impaired, contributing to lower academic achievements in this population [26].

This study has notable limitations that must be acknowledged. First, its cross-sectional design prevents the establishment of causality. Further investigation is needed to determine whether decreasing UPF consumption results in superior academic performance. Furthermore, the relationship’s direction is uncertain, as higher academic performance might explain lower UPF consumption. To ascertain if increased UPF consumption directly leads to poorer academic results in adolescents, longitudinal investigations are necessary. Second, the reliance on self-reported data introduces potential social desirability and recall biases, possibly influencing the accuracy of the reported UPF consumption. Third, while the analysis considers various covariates, academic performance is affected by multiple factors and unaccounted variables could impact the observed outcomes. Fourth, the findings may not be fully applicable to other regions of Spain or different countries due to contextual and cultural variations in dietary habits. Lastly, while the biological mechanisms linking the consumption of UPFs with academic performance are likely universal, external factors such as access to healthy foods, cultural traditions like the practice of sharing family dinners, variations in UPF composition resulting from local regulations, and socioeconomic status can significantly influence the magnitude and direction of these effects. Despite these limitations, the study provides cross-sectional evidence on the association of UPF consumption with academic performance among adolescents, an understudied group. The inclusion of numerous covariates, including lifestyle, sociodemographic, and anthropometric variables, strengthens the consistency of the findings.

## 5. Conclusions

Our study identifies a negative association between UPF consumption and academic performance in adolescents, highlighting it as a modifiable factor that could impact academic outcomes. Adolescents with higher UPF consumption exhibited consistently lower grades across various academic indicators, emphasizing the importance of dietary quality during this critical developmental period. This finding highlights the need for a multifaceted approach to promote healthier eating habits that involves schools, families, and public health policies. Implementing strategies such as nutrition education programs tailored to adolescents, enhancing the nutritional standards of school meal offerings, and restricting the availability of UPFs in educational environments (e.g., through policies that limit the sale of UPFs in vending machines, cafeterias, and school events or by providing healthier, more appealing alternatives) could significantly contribute to improving academic outcomes. Furthermore, these interventions have the potential to mitigate disparities in educational attainment that are linked to diet quality. Given the long-term psychological, social, and economic implications of academic achievement, reducing UPF consumption among adolescents should be prioritized as part of broader public health efforts aimed at supporting youth development and success.

## Figures and Tables

**Figure 1 nutrients-17-00524-f001:**
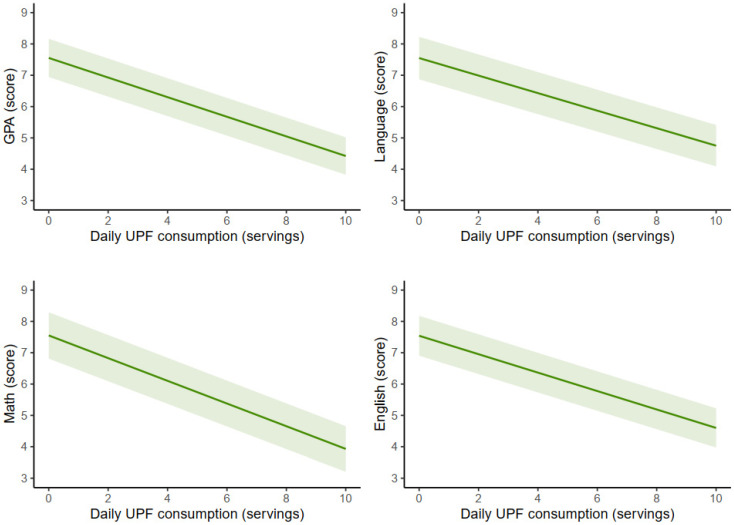
Estimated marginal means of different academic performance indicators based the daily servings of ultra-processed food consumed among adolescents. GPA, grade point average (i.e., the mean of all academic subject scores for each student); UPF, ultra-processed food consumption. Adjusted for sex, age, socioeconomic status, sedentary behavior, physical activity, sleep duration, and body mass index.

**Table 1 nutrients-17-00524-t001:** Characteristics of the adolescents included in the study.

Variable	*N* = 788 ^1^
Age (years)	14.0 (13.0, 15.0)
Sex	
Boys	352 (44.7%)
Girls	436 (55.3%)
FAS-III (score)	8.0 (7.0, 10.0)
BMI (kg/m^2^)	21.6 (19.3, 25.3)
YAP-S sedentary behaviors (score)	2.6 (2.2, 3.0)
YAP-S physical activity (score)	2.6 (2.2, 3.0)
Overall sleep duration (minutes)	497.1 (458.6, 527.1)
Energy intake (kcal)	2589.7 (1963.6, 3440.8)
UFP (g)	235.0 (151.1, 380.7)
UFP (servings)	4.0 (2.9, 6.0)
GPA (score)	6.3 (4.7, 7.7)
Language (score)	7.0 (5.0, 8.0)
Maths (score)	6.0 (4.0, 8.0)
Foreign language (score)	6.0 (5.0, 8.0)

^1^ Median (interquartile range) or number (percentage).

**Table 2 nutrients-17-00524-t002:** Association of ultra-processed food consumption status with academic performance among adolescents.

	GPA	Language	Maths	English
UPF Consumption Status	M (95% CI)	M (95% CI)	M (95% CI)	M (95% CI)
Low	6.6 (6.4 to 6.9) ^a^	7.0 (6.7 to 7.3) ^a^	6.2 (5.9 to 6.5) ^a^	6.6 (6.4 to 6.9) ^a^
Medium	6.4 (6.1 to 6.7) ^a^	6.7 (6.4 to 7.0) ^a^	6.0 (5.7 to 6.4) ^a^	6.6 (6.3 to 6.9) ^a^
High	5.6 (5.4 to 5.9)	6.0 (5.7 to 6.3)	5.2 (4.9 to 5.6)	5.7 (5.5 to 6.0)

Adjusted for sex, age, socioeconomic status, sedentary behaviors, physical activity, sleep duration, and body mass index. CI, confidence interval; GPA, grade point average; M, mean; UPF, ultra-processed food. ^a^ Significant difference from high UPF consumption (*p* < 0.001).

## Data Availability

The original contributions presented in this study are included in the article/Appendix A. Further inquiries can be directed to the corresponding author.

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
