# Peer review of "Investigating the Relationship Between Ultra-Processed Food Consumption and Academic Performance in the Adolescent Population: The EHDLA Study"

_nutrients, 2025, doi:10.3390/nu17030524_

Round 1

Reviewer 1 Report

Comments and Suggestions for Authors

Thank you for the opportunity to review this study entitled “Investigating the relationship between ultra-processed food consumption and academic performance in the adolescent population: The EHDLA study” (nutrients-3459854).

The aim of the present manuscript is to explore the relationship between ultra-processed food (UPF) consumption and academic performance in adolescents. The research involved a sample of 788 Spanish adolescents (aged 12-17).

In my view, the research topic is highly relevant, and the study is engaging. The specificity of the sample and the unique emphasis on UPF are key strengths of the paper. However, there are several aspects that need attention before the manuscript is ready for publication.

·       abstract: Kindly include details about the sample (e.g., mean age, standard deviation, and gender distribution) to provide a clearer overview of the study's content.

·       Abstract: Avoid including statistical indices such as p-values, confidence intervals, etc., in this section.

·       Keywords: Ensure the keywords are arranged in alphabetical order.

·       Introduction: The research gap is only briefly addressed at the end of the introduction. Consider elaborating on it further to emphasize the importance of this study.

·       Measures: Please calculate and report the internal consistency of the self-report scales used in this sample.

·       Results and Discussion: These sections are well-organized and do not raise any concerns.

·       Limitations: Expand the suggestions for future research to provide more depth.

·       Conclusions: Further elaborate on the practical implications of the findings to strengthen this section.

Author Response

Thank you very much for your comments and recommendations. These have been considered and included in the current version of the manuscript. For more information, please see the attached file.

Reviewer 2 Report

Comments and Suggestions for Authors

The article addresses a relevant and timely topic, exploring the relationship between ultra-processed food (UPF) consumption and academic performance in adolescents. The methodological design, detailed statistical analysis, and well-founded discussion reflect rigorous research efforts. However, to strengthen the manuscript and maximize its impact, several areas for improvement have been identified, as outlined below.

Strengths of the Article

1.     Relevance of the topic:

o   The relationship between diet and academic performance is of great interest to researchers, policymakers, and families. The focus on adolescents, a population particularly vulnerable to dietary and environmental influences, is especially significant.

2.     Methodological robustness:

o   The use of generalized linear models (GLMs) to control for potential covariates is appropriate and robust.

o   Including multiple indicators of academic performance (GPA, Language, Math, and English) provides a comprehensive perspective on the findings.

o   The analysis by tertiles of UPF consumption adds clarity and practicality to the interpretation of the data.

3.     Contribution to the field:

o   The results complement prior research by identifying a significant negative relationship between UPF consumption and academic performance, highlighting potential physiological and behavioral mechanisms.

o   The discussion addresses relevant hypotheses and supports the findings with up-to-date literature.

Areas for Improvement and Recommendations

1.     Clarity in objectives and study rationale:

o   Recommendation: Reframe the study objectives in the introduction to make them more specific and aligned with the findings. For instance, instead of presenting a general relationship between UPF consumption and academic performance, emphasize the intent to analyze effects on specific subjects.

2.     Methodology description:

o   Although the methodology is solid, certain aspects could be better detailed:

§  Sample size: Provide an explicit justification for the sample size used and indicate whether a power calculation was conducted.

§  Definition of UPFs: Elaborate on the classification of ultra-processed foods and specify whether a validated tool (e.g., the NOVA system) was employed.

§  Measurement instruments: Clarify the method used to assess academic performance (e.g., were data self-reported or obtained from school records?).

3.     Statistical analysis:

o   While the covariates considered are mentioned, the criteria for their selection could be further explained. Additionally:

§  Recommendation: Assess whether the statistical model includes potential interactions between covariates (e.g., gender or socioeconomic status) and UPF consumption.

§  Supplementary results: Consider presenting additional analyses, such as sensitivity analyses or evaluations of the impact of specific types of UPFs (e.g., sugary beverages vs. snacks).

4.     Discussion and physiological mechanisms:

o   The discussion of potential biological mechanisms is solid but could be enriched in certain areas:

§  Human vs. animal evidence: Highlight the limitations of extrapolating findings from animal models to human adolescents.

§  Recommendation: Include more references to longitudinal or experimental studies in humans, if available, to better contextualize the findings.

§  Practical implications: Further explore how these mechanisms might vary based on sociodemographic or cultural factors.

5.     Limitations and generalizability:

o   Although the limitations are well described, certain areas warrant additional emphasis:

§  Directionality of the relationship: While the cross-sectional nature of the study is acknowledged, provide more in-depth discussion on how future research could address causality, such as through longitudinal studies or targeted interventions.

§  Generalizability: Highlight that the findings reflect a specific context and that cultural or educational differences might influence the observed relationship.

6.     Relevance of practical implications:

o   While the practical implications are appropriate, they could be more specific:

§  Recommendation: Provide concrete examples of school-based interventions or public health policies that could be implemented, such as strategies to limit the availability of UPFs in educational settings.

7.     Presentation of results:

o   Tables and figures: While clear, their presentation could be improved:

§  Ensure that the figures are self-explanatory, meaning they include sufficient information to be understood independently of the main text.

§  Consider adding visual representations of the differences between UPF consumption tertiles and their association with each academic indicator.

Report Conclusion:

This article represents a valuable contribution to the field of nutrition and academic performance in adolescents. However, incorporating the suggestions outlined above could further enhance the manuscript, particularly in terms of clarity, methodological rigor, and practical relevance.

These improvements would not only enrich the scientific impact of the study but also increase its applicability for policymakers, educators, and health professionals.

We remain available to discuss any aspect of the report or collaborate on necessary revisions.

Author Response

(The authors gave the same response as above.)

Round 2

Reviewer 2 Report

Comments and Suggestions for Authors

The new version is ok